# Pinocembrin Ameliorates Skin Fibrosis via Inhibiting TGF-β1 Signaling Pathway

**DOI:** 10.3390/biom11081240

**Published:** 2021-08-19

**Authors:** Xiaohe Li, Yunqian Zhai, Buri Xi, Wei Ma, Jianwei Zhang, Xiaoyang Ma, Yang Miao, Yongjian Zhao, Wen Ning, Honggang Zhou, Cheng Yang

**Affiliations:** 1State Key Laboratory of Medicinal Chemical Biology, College of Pharmacy, College of Life Sciences, Nankai University, Tianjin 300353, China; lixiaohe908@163.com (X.L.); zhaiyunqian1996@163.com (Y.Z.); xiburige3222@163.com (B.X.); jianweizhang2021@163.com (J.Z.); xiaoyangma1014@163.com (X.M.); miaoyangtcm@163.com (Y.M.); yangcheng@nankai.edu.cn (C.Y.); 2Tianjin Key Laboratory of Molecular Drug Research, Tianjin International Joint Academy of Biomedicine, Tianjin 300457, China; 3Department of Burn and Plastic Surgery, Tianjin Fourth Hospital, Nankai University, Tianjin 300322, China; mwtyq6000@163.com (W.M.); zhyj07-love@163.com (Y.Z.)

**Keywords:** skin fibrosis, keloid, fibroblasts, TGF-β1 signaling, pinocembrin

## Abstract

Skin fibrotic diseases, such as keloids, are mainly caused by pathologic scarring of wounds during healing and characterized by benign cutaneous overgrowths of dermal fibroblasts. Current surgical and therapeutic modalities of skin fibrosis are unsatisfactory. Pinocembrin, a natural flavonoid, has been shown to possess a vast range of pharmacological activities including antimicrobial, antioxidant, anti-inflammatory, and anti-tumor activities. In this study we explored the potential effect and mechanisms of pinocembrin on skin fibrosis in vitro and in vivo. In vitro studies indicated that pinocembrin dose-dependently suppressed proliferation, migration, and invasion of keloid fibroblasts and mouse primary dermal fibroblasts. The in vivo studies showed that pinocembrin could effectively alleviate bleomycin (BLM)-induced skin fibrosis and reduce the gross weight and fibrosis-related protein expression of keloid tissues in xenograft mice. Further mechanism studies indicated that pinocembrin could suppress TGF-β1/Smad signaling and attenuate TGF-β1-induced activation of skin fibroblasts. In conclusion, our results demonstrate the therapeutic potential of pinocembrin for skin fibrosis.

## 1. Introduction

Skin fibrosis is a pathological result of abnormal tissue repair after deep skin dermis injury and occurs in a variety of pathological processes, such as immune disease scleroderma (SSc), keloids, and hypertrophic scars (HS) [1,2]. For instance, keloids are a skin fibrotic disease characterized by excessive deposition of extracellular matrix such as collagen in the dermis and subcutaneous tissues caused by human skin injuries, burns, or surgery [3]. Keloids are similar to benign tumor lesions, which show tumor-like hyperplasia, exceed the boundaries of the original injury, and invade adjacent tissues to cause dysfunction and bring damage to the patient’s appearance and limb function [4]. Keloid scars easily relapse after simple surgical resection, and the side effects of therapeutic drugs are relatively numerous [5]. Therefore, it is of great necessity to find novel therapeutic targets and develop effective treatment drugs.

Recent studies have shown that excessive proliferation of fibroblasts, extracellular matrix deposition, and inflammatory cell infiltration in the process of tissue repair together constitute the biological basis of keloid formation [6]. Among them, fibroblasts play a key role in tissue healing and scar formation [7]. Although the specific pathogenic mechanism of keloid has not been clearly studied, existing research shows that TGF-β1 is the most critical regulatory factor among many cell growth factors related to the activation of pathological scar fibroblasts [8,9,10]. TGF-β1 can stimulate fibroblast activation, proliferation, and apoptosis resistance by inducing the downstream Smad signal pathway. TGF-β1 can also promote the synthesis and deposition of extracellular matrix such as type I and type III collagen, inhibit collagenase activity, and cause abnormal collagen degradation, thereby promoting continuous growth of keloids [11,12,13]. Therefore, TGF-β/Smads signaling has become a potential therapeutic target for skin fibrotic diseases.

Pinocembrin (5,7-dihydroxyflavonone) is the flavonoid compound with the highest content in propolis and has various biological functions such as anti-inflammatory, anti-oxidation, and anti-apoptosis [14]. The anti-fibrosis effects of pinocembrin have also been reported. Studies have shown that pinocembrin could protect cardiac fibrosis and arrhythmia caused by myocardial ischemic injury (I/R) through its anti-inflammatory and anti-oxidant activity [15]. Other studies also show that pinocembrin could alleviate CCl4-induced chronic liver fibrosis by inhibiting the TGF-β1/Smad signaling pathway [16]. However, there were no reports on the role and mechanism of pinocembrin in skin fibrosis. In this study, we performed in vitro and in vivo experimental models to evaluate the anti-fibrotic effects and possible mechanisms of pinocembrin on skin fibrosis.

## 2. Materials and Methods

### 2.1. Human Keloid Tissue Samples

Keloid tissues were collected from three patients during keloid removal surgery in Tianjin Fourth Hospital Affiliated with Nankai University. All human participants in this study signed informed consent before enrolling in the study, and all procedures in this study were approved by the Ethics Committee of Nankai University on 30 November 2020 (approval No. NKUIRB2020064). All experimental methods involving human subjects were completed in accordance with the relevant guidelines and regulations.

### 2.2. Separation and Culture of Keloid Fibroblasts

Separation and culture of keloid fibroblasts (KFs) were performed according to previous protocol [17]. In brief, tissue specimens from keloids were cut under aseptic conditions, and epithelial and subcutaneous tissues were removed carefully with ophthalmic scissors. Then, the specimens were cut into tissue pieces of 5 mm^3^, placed in DMEM medium containing 10% FBS, and cultivated in a 5% CO_2_ incubator at 37 °C. When the cell growth was close to confluence, the cells were passaged at a proportion of 1:3.

### 2.3. Separation and Culture of Primary Dermal Fibroblasts

The primary dermal fibroblasts (DFs) from newborn mouse dermises were separated according to the methods as previously described [18]. The cells were inoculated in DMEM medium containing 10% FBS and cultured at 37 °C in a humidified 5% CO_2_ atmosphere.

### 2.4. Cell Counting Kit-8 (CCK-8) Assay

The drug effect on the proliferation of KFs was determined by CCK-8 (Solarbio). A total of 1000 primary cultured KFs were seeded in 96-well plates per well and exposed to pinocembrin (0 to 80 µM). The cells were tested with CCK-8 reagent at days 1, 3, 5, 7, and 9. The medium optical density value of each well was measured at 450 nm by a microplate reader.

### 2.5. EdU Incorporation Assay

Cell proliferation was detected using the BeyoClick^TM^ EdU-555 Imaging Kit (Beyotime Biotechnology, Shanghai, China). After treatment with different concentrations of pinocembrin for 24 h, DFs were incubated with EdU for 2 h, fixed in 4% paraformaldehyde (PFA), and incubated with a Click Additive Solution for 30 min. Immunofluorescence signals were captured using a laser scanning confocal microscope (Leica, TCS SP8, Wetzlar, Germany).

### 2.6. Wound-Healing Assay

KFs or DFs were seeded in a twelve-well plate, and the cell monolayer was scratched using a sterile 200 µL pipette tip. The cells were then treated with pinocembrin in the presence or absence of TGF-β1 (5 ng·mL^−1^). The scratch was observed at 0, 12, 24, 36, and 48 h by an inverted optical microscope, and each group was imaged at three different locations. Images were obtained for analysis using Image J software.

### 2.7. Transwell Assays

Transwell chambers (8-μM pore size; Corning, New York, NY, United States) were used to confirm the in vitro anti-migratory and anti-invasive effect of pinocembrin. For the transwell migration and invasion assays, the upper chamber was coated with or without Matrigel (BD Biosciences, New York, NY, USA). Subsequently, 3 × 10^4^ KFs or DFs in the serum-free medium was seeded in the upper chamber, and various concentrations of pinocembrin were added. The lower chamber was supplemented with DMEM medium containing 15% FBS and the same concentration of pinocembrin as that in the upper compartment. After incubation for 24–48 h, the migrated cells were fixed with 4% paraformaldehyde, stained with DAPI (Beyotime Biotechnology, Shanghai, China), and imaged with the microscope.

### 2.8. Ex Vivo Explant Culture of Human Keloid Tissues

Keloid tissues removed from the patients were cut into 3 × 3 × 2 mm fragments, seeded onto culture dishes, and incubated in DMEM containing 10% FBS. Once the tissue fragments adhered to the bottom of the dish, the medium was replaced with or without various pinocembrin concentrations. Representative images were acquired on day 9 after the KFs migrated from the edge of the tissue.

### 2.9. Quantitative Real-Time PCR (qRT-PCR)

In order to obtain gene expression of certain proteins in cells and tissues, the total RNA was isolated with TRIzol Reagent and reverse-transcribed with FastKing gDNA Dispelling RT SuperMix (TIANGEN Biotech, Beijing, China). QRT-PCR was performed with UNICON^®^ qPCR SYBR Green Master Mix (Yeasen Biotech, Shanghai, China) according to the manufacturer’s protocols. The sequences of primers were listed in Table 1. GAPDH or β-Actin was used as the endogenous reference gene in all RT-PCR experiments. The quantification of gene expression was performed relative to an endogenous reference gene (GAPDH/β-actin) using the −△△CT method in the experiments [19].
△CT = CT gene of interest − CT endogenous reference gene(1)
△△CT = △CT treatment group − △CT control (CTL) group(2)

### 2.10. Western Blot Analysis

All proteins were collected from cells or tissues as previously described [18]. The total protein samples were separated by SDS-PAGE gel electrophoresis and transferred to PVDF membrane. After blocking, the immunoblots were probed with the following primary antibodies: α-SMA, Collagen I, Fibronectin, Phospho-Smad2, Smad2, Phospho-Smad3, Smad3, β-tubulin, and GAPDH. Then, the membranes were incubated with HRP-conjugated secondary antibodies and detected with an ECL system (Affinity Bioscience, Cincinnati, OH, United States).

### 2.11. Immunofluorescence Staining

Cells were fixed with 4% PFA for 20 min, permeabilized with 0.2% Triton X-100 for 10 min, and blocked with 5% BSA for 1 h. Cells were incubated with rabbit anti-α-SMA (1:200) overnight at 4 °C, followed by FITC-conjugated secondary antibody. Cells were counterstained with DAPI. Fluorescence was analyzed using a laser scanning confocal microscope (Leica, TCS SP8, Wetzlar, Germany).

Skin tissues were embedded in OCT compound and sectioned on a cryostat (5 μm thick). After fixation in PFA, sections were blocked with 5% BSA for 1 h and then incubated with FITC-anti-Phospho-Smad2 (1:100) or FITC-anti-Phospho-Smad3 (1:100) overnight at 4 °C. The sections were counterstained with DAPI. Fluorescence was captured using a Leica TCS SP8 confocal microscope.

### 2.12. Animals

C57BL/6 mice (male, 6–8 weeks) and nude BALB/c mice (female, 10 weeks) were acquired from the Laboratory Animal Center, Academy of Military Medical Sciences of People’s Liberation Army (Beijing, China). Animal experiments were approved by the Institutional Animal Care and Use Committee (IACUC) of Nankai University (approval No. SYXK 2014-0003), and all methods were performed in accordance with the relevant guidelines and regulations.

### 2.13. Bleomycin-Induced Skin Fibrosis Model

Skin fibrosis was induced by intradermal injection of bleomycin (BLM) sulfate (100 μL, 500 µg/mL in vehicle) into a single location on the shaved back skin of C57BL/6 mice once daily for 3 weeks. A total of 40 mice were equally randomized into five groups: (1) NaCl group, where mice were injected with vehicle (100 μL sterile saline plus 1% DMSO) intradermally; (2) BLM group, where mice were treated with 100 μL BLM sulfate intradermally; (3–5) treatment groups, where mice were treated with 100 μL BLM plus pinocembrin (20, 40, 80 μM) intradermally.

### 2.14. Keloid Xenograft Mouse Model

The keloid xenograft mouse model was performed as previously described [20]. Briefly, the keloid tissues were cut into approximately 5 ×  5  ×  5 mm sections, and the weight was ranged from 0.08 to 0.1 g per tissue block. The nude BALB/c mice were anesthetized, a cut approximately 0.5 cm was made on the back of the mice, a subcutaneous dissection was made to form a pocket, and a tissue block was implanted into the subcutaneous pocket. After surgery, the keloid xenograft model was established at around 14 days. The transplants on each mouse were assigned for injection with vehicle (phosphate buffer solution) or pinocembrin (20, 40, 80 µM) for six treatments over two weeks. The weight of the transplants was evaluated, and further analyses of histology and gene expressions were performed.

### 2.15. Histological Examination

Skin samples were fixed in 10% formalin, dehydrated, embedded in paraffin, and cut into sections 5 µm thick. After deparaffinizing in xylene and rehydrating using an alcohol series, the sections were stained with hematoxylin and eosin (H&E), Masson’s trichrome, and Picrosirius red (Solarbio, Beijing, China). Images were randomly photographed with an upright transmission fluorescence microscope (Olympus, Tokyo, Japan) and analyzed by Image-Pro Plus Version 6.0.

### 2.16. Hydroxyproline Content Determination

Skin (5 mg) of mice was isolated, placed in ampoules for drying, and combined with hydrochloric acid. Then, the mixture was adjusted to pH 6.5–8.0, filtered, and adjusted to a total volume of 5 mL with 1× PBS. Hydroxyproline (HYP) was tested by a hydroxyproline content detection kit, and the absorbance was measured at 577 nm.

### 2.17. Flow Cytometric Analysis

Flow cytometric analysis was performed using an annexinV-FITC apoptosis detection kit (Beyotime, Shanghai, China) as described by the manufacturer’s instructions. The mouse primary dermal fibroblasts/mL (1 × 10^6^) were seeded into six-well plates and left for 24 h in an incubator to resume exponential growth. Cells were treated with indicated doses of pinocembrin (20 μM, 40 μM, 80 μM) for 48 h. Then, they were collected and washed with PBS twice and gently resuspended in annexin-V binding buffer and incubated with annexinV-FITC/PI in the dark for 15 min, then tested on a flow cytometer (BD LSRFortessa).

### 2.18. Molecular Docking

The crystal structure (PDB ID: 2WOT) was extracted from the RCSB Protein Data Bank (PDB). The protein structure was prepared using the Protein Preparation Wizard module in Schrodinger 2017 to remove all crystallographic water molecules, correct side chains with missing atoms, add hydrogen atoms and assign protonation states and partial charges with the OPLS_2005 force field. After that, the protein structure was minimized until the root-mean-square deviation (RMSD) of the nonhydrogen atoms reached less than 0.3 Å. The structure of pinocembrin was prepared using the LigPrep module of the Schrodinger 2017 molecular modeling package to add hydrogen atoms, convert 2D structures to 3D, generate stereoisomers, and determine the ionization state at pH 7.0 ± 2.0 with Epik. Using the prepared receptor structure, a receptor grid was generated around the original ligand site of the crystal structure. Then, the pinocembrin was docked to the receptor using the Glide XP protocol.

### 2.19. Statistical Analysis

Data were presented using the Prism version 7.0 software as the means ± SD. Differences between experimental and control groups were assessed by Student’s *t* test. Significant differences among multiple groups were detected by one-way ANOVA. *p* < 0.05 was considered statistically significant.

## 3. Results

### 3.1. Pinocembrin Inhibits TGF-β1-Induced Proliferation and Migration of Mouse Primary Dermal Fibroblasts

First, we measured the toxicity effect of pinocembrin on the mouse primary dermal fibroblasts (DFs). Results of MTT experiments showed that the IC50 value of pinocembrin in DFs was between 336.9 μM and 380.6 μM (Appendix A). At the same time, we referred to the effective concentration range of pinocembrin in the previous studies [21,22], and we selected three concentrations of 20, 40, and 80 μM for follow-up pharmacological experiments. We also tested the effect of pinocembrin on the apoptosis and necrosis of DFs under these three concentration gradients. Results showed that the effect of pinocembrin on normal cell apoptosis and necrosis was not significant (Appendix A).

Next, we measured the effect of pinocembrin on TGF-β1-induced proliferation and migration of DFs. The MTT experiments and the cell counting results indicated that after 24 h of growth, the cell viability and numbers were significantly increased, and the cell viability and numbers were more increased after stimulation with TGF-β1 (Appendix A). After treatment with different doses of pinocembrin (20 μM, 40 μM, 80 μM), the cell viability of control groups did not change significantly, while the cell viability in TGF-β1 co-treated groups was obviously inhibited by pinocembrin, which indicated that pinocembrin could suppress the proliferation of activated skin fibroblasts (Appendix A). Similarly, pinocembrin could also significantly inhibit the increase in the number of DFs induced by TGF-β1 at 48 h (Appendix A). We also used the EdU incorporation assay to evaluate the effect of pinocembrin on proliferation of active DFs. As presented in Figure 1A,B, in vitro growth of cultured mouse primary dermal fibroblasts (DFs) accelerated in response to the stimulation of TGF-β1, while cell proliferation was suppressed after pinocembrin treatment, as the EdU-positive cell percentage dropped significantly in the EdU incorporation assay.

In addition, the scratch assay demonstrated a significant inhibitory effect of pinocembrin on the migration of TGF-β1-activated dermal fibroblasts compared to that in nontreated cells (Figure 1C). At both time points, quantitative analysis further confirmed a significant dose-dependent reduction in cell migration in the treated groups (Figure 1D). Meanwhile, the transwell assay showed that the numbers of migratory cells in the TGF-β1-treated group were increased compared with the control group, while they were dramatically decreased after intervention with pinocembrin (Figure 1E,F).

### 3.2. Pinocembrin Suppresses the Proliferation, Migration, and Invasion of Keloid Fibroblasts

Since excessive and abnormal proliferation of fibroblasts has been reported in keloids, we evaluated the regulatory function of pinocembrin on the proliferation of keloid fibroblasts (KFs) by using the CCK-8 assay. In the pinocembrin-treated groups, cell proliferation was dose-dependently inhibited during the 9-day time period compared to the control group (Figure 2A).

Next, we used wound healing assays to test the effects of pinocembrin on the migration of KFs. The experimental results showed that pinocembrin was able to inhibit KF migration markedly (Figure 2B,C), and the ratio of migrated cells to total cells also indicated the same conclusion (Appendix A). At the same time, the transwell assay also revealed that pinocembrin inhibited the migration of KFs by reducing the number of cells that migrated to the bottom surface of the upper chamber (Figure 2D,E). To further investigate the inhibitory effect of pinocembrin on KF migration, we established an ex vivo keloid explant culture model. KFs migrated out of the edges of the cultured keloid explants within two weeks, while culturing with pinocembrin significantly inhibited KF migration in a dose-dependent manner (Figure 2F,G).

Moreover, we investigated the effect of pinocembrin on the invasive capability of KFs by transwell invasion assay. Not surprisingly, fewer cells traversed the Matrigel-coated polycarbonate membrane in the pinocembrin-treated groups, which revealed pinocembrin diminished the invasive capability of KFs (Figure 2H,I).

### 3.3. Pinocembrin Attenuates TGF-β1-Induced Activation of Mouse Primary Dermal Fibroblasts

Using an in vitro cell culture model of treating mouse primary dermal fibroblasts with TGF-β1 and different doses of pinocembrin, the drug effects on the expression of fibrotic factors were investigated. As presented in Figure 3A, pinocembrin significantly reversed the stimulatory effect of TGF-β1 on α-SMA, Col1, and Fn protein levels in a dose-dependent manner. Consistent with the protein expression results, treatment with pinocembrin significantly suppressed the gene expression of α-SMA, Col1, and Fn (Figure 3B). The result was also confirmed by immunofluorescence assay, which showed decreasing expression of α-SMA with increasing drug concentrations (Figure 3C). Interestingly, the inhibitory effect of pinocembrin was also supported by decreased gene expression of matrix metalloproteinases (MMPs). As shown in Figure 4B, MMP-2, MMP-9, and MMP-14 gene levels were significantly reduced in the drug-treated groups. Further study showed that the phosphorylation level of Smad2/3, which is strongly increased by TGF-β1, was significantly down-regulated in the pinocembrin-treated groups, while there was no significant influence on the protein levels of total Smad2/3 (Figure 3D). Accordingly, these results revealed that pinocembrin could suppress TGF-β1-induced fibroblast differentiation, thus reducing ECM production.

### 3.4. Pinocembrin Reduces Fibrosis-Associated Expression of Keloid Pathogenesis in Keloid Fibroblasts

We next studied if pinocembrin directly affected the profibrotic phenotype of keloid fibroblasts in vitro. To evaluate the molecular mechanism, we mainly detected the expression of α-smooth muscle actin (α-SMA), which is a typical fibroblast activation marker, as well as collagen I (Col1) and fibronectin (Fn), which reflect extracellular matrix (ECM) deposition. As shown in Figure 4A, pinocembrin treatment dose-dependently decreased the protein expression of α-SMA, Col1, and Fn. Similarly, qPCR analysis demonstrated that pinocembrin significantly down-regulated the mRNA expression levels of a-SMA, Col1, and Fn (Figure 4B). Meanwhile, immunofluorescence assays of α-SMA showed the same conclusion (Figure 4C).

Whereas the TGF-β signaling pathway exerts an important role in the process of skin fibrosis, we evaluated the effect of pinocembrin on the activation of Smad2/3 of TGF-β downstream. As shown in Figure 4D, pinocembrin significantly inhibited the phosphorylation of Smad2 and Smad3 in a concentration-dependent manner compared with the control group. Overall, these consequences indicated that pinocembrin reduces fibrosis-associated expression of keloid pathogenesis at least partly through the TGF-β signal pathway in keloid fibroblasts.

### 3.5. Pinocembrin Alleviates BLM-Induced Skin Fibrosis in Mice and Inhibits Fibrogenic Activation In Vivo

To determine the effect of pinocembrin on BLM-induced skin fibrosis, C57BL/6 mice were subcutaneously co-administered both BLM and pinocembrin once daily for 3 weeks. Compared to the model group, pinocembrin significantly decreased dermal thickness and collagen deposition in BLM-induced mice, as demonstrated by H&E, Masson’s trichrome, and Sirius red staining (Figure 5A–C). Consistently, pinocembrin significantly reduced the BLM-increased hydroxyproline content, which is an indicator of the severity of fibrosis in skin (Figure 5D). Meanwhile, pinocembrin treatment reduced protein levels of α-SMA and Col1 (Figure 5E). The result was further supported by the lower mRNA expressions of α-SMA, Col1a1, Col1a2, and Fn (Figure 5F). In addition, RT-PCR assay also showed that pinocembrin down-regulated the mRNA of MMP-2, MMP-9, and MMP-14 in skin tissue (Figure 5F). Moreover, immunofluorescence data revealed that pinocembrin downregulated the level of p-Smad3 in skin frozen sections of the BLM-induced model in a dose-dependent manner (Figure 5G). These findings suggest that pinocembrin exerts a potent anti-fibrotic effect on skin fibrosis by reducing the production of collagen and regulating expression of various soluble factors in the BLM-induced model in mice.

### 3.6. Intralesional Injection of Pinocembrin Reduces the Size and ECM Gene Expression of Xenografted Keloid Tissue

To evaluate the effect of pinocembrin in a humanized animal model, we established a keloid xenograft nude mouse model. The model is comparable to studying the specific human tissue in an in vivo environment because the complex human cell–cell interactions in both the epidermis and dermis are preserved. Following six intralesional injections with and without pinocembrin for two weeks, the transplanted keloid tissues were excised for analysis (Figure 6A). The weights of pinocembrin-treated tissues were lighter than those of the vehicle group (Figure 6B). qPCR analysis revealed that pinocembrin treatment dose-dependently decreased the expression of α-SMA, Col1α1, Col3α1, and Fn in keloid xenograft tissues (Figure 6C). Collectively, these results suggested that pinocembrin significantly accelerated the regression of xenografted keloid tissues by reducing gene expression of the extracellular matrix.

## 4. Discussion

Skin fibrosis is the histopathologic hallmark of dermatologic disease and is characterized by excessive proliferation of skin fibroblasts and deposition of extracellular matrix [23]. Keloids are one of the skin fibrotic diseases difficult to treat, and they easily recur [24]. Although there are many studies on the disease mechanism of keloids, there are very few interventions and treatments that have been put into clinical trials [25]. In this study, we explored the potential effects and mechanisms of pinocembrin on skin fibrosis in vivo and in vitro. The in vitro experiments showed that pinocembrin could inhibit the proliferation, migration, invasion, and activation of skin fibroblasts and reduce the deposition of extracellular matrix. In vivo studies showed that pinocembrin could effectively reduce bleomycin-induced skin fibrosis, and it can also significantly reduce the weight of scar tissue and the expression level of fibrosis-related proteins in keloid xenograft mice.

In the mechanism study, we found that pinocembrin can inhibit TGF-β1/Smad signal transduction in a dose-dependent manner, and it can attenuate the activation of fibroblasts induced by TGF-β1. Emerging evidence has shown that flavonoids with 5, 7, 3′, 4′ hydroxyl substitutions could bind to activin receptor-like kinase 5 (ALK5) to down-regulate the signal transduction of TGF-β/Smads and reduce skin fibrosis [26]. Pinocembrin is a flavonoid with 5 and 7 hydroxyl substitutions, and preliminary experiments showed that pinocembrin may bind to ALK5 (Appendix A); thus, the specific target of pinocembrin in skin fibrosis may also be related to ALK5, which needs further study.

In this study, we established a keloid xenotransplantation model in nude mice to simulate the incidence of scarring in the human body. Studies have shown that keloid xenon transplants have an angiogenic and continuous ability to synthesize collagen and can remain active for weeks to months after transplantation in thymus-free nude mice; thus, the model can be compared to studying specific human tissue in the in vivo environment [27,28]. However, this model still has its limitations. Although thymus-free nude mice show limited transplant rejection due to lack of functional T-cells, thymus-free naked mice do not completely lack an immune system; they still maintain a functioning congenital and bodily fluid adaptive immune system, including functional killer cells. This may affect the vitality of the transplant tissue, limiting the overall effectiveness of the model [29]. Our results showed that intralesional injection of pinocembrin could reduce the size and extracellular matrix gene expression of xenografted keloid tissue, which was in line with the in vitro and animal model results.

## 5. Conclusions

In summary, our study indicated that pinocembrin could suppress activation of keloid fibroblasts and mouse dermal fibroblasts via inhibiting TGF-β1/Smad signaling, and it could effectively alleviate skin fibrosis in a mice model and keloid xenografts. Based on the above results, we believe that pinocembrin may serve as an anti-fibrotic drug candidate in skin fibrosis treatment. Therefore, as a traditional Chinese medicine monomer, pinocembrin can not only be developed as a drug for the treatment of skin fibrosis, but the mother nucleus of pinocembrin can also be used as the basis for the development of anti-skin fibrosis drugs, which provides more optional drug candidates for skin fibrosis such as keloids and other fibrotic diseases.

## Figures and Tables

**Figure 1 biomolecules-11-01240-f001:**
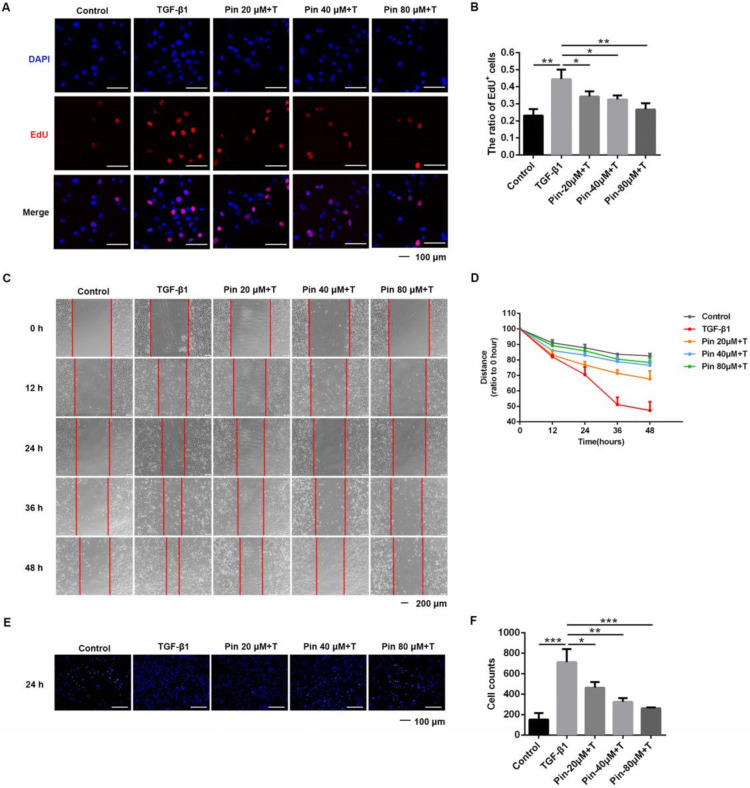
Pinocembrin suppresses TGF-β1-induced proliferation and migration of mouse primary dermal fibroblasts. (**A**,**B**) Representative immunofluorescence images and quantitative results of EdU incorporation assay in DFs co-cultured with TGF-β1 (5 ng·mL^−1^) and pinocembrin (0, 20, 40, 80 μM) (×40, scale bar = 100 μm). The ratio of EdU-positive cells to DAPI-labeled cells in each group was determined. (**C**,**D**) Images and quantitative analysis of wound healing assay in DFs treated with TGF-β1 (5 ng·mL^−1^) and different doses of pinocembrin. The wound closure was captured at 0, 12, 24, 36, and 48 h after scratch generation. (**E**,**F**) The inhibitory effect of pinocembrin on the TGF-β1-induced migration of DFs was investigated by the transwell assay. Images were captured and counted under a fluorescence microscope at ×200 (scale bar = 100 μm). Data were presented as the means ± SD, n = 3. * *p* < 0.05; ** *p* < 0.01; *** *p* < 0.001.

**Figure 2 biomolecules-11-01240-f002:**
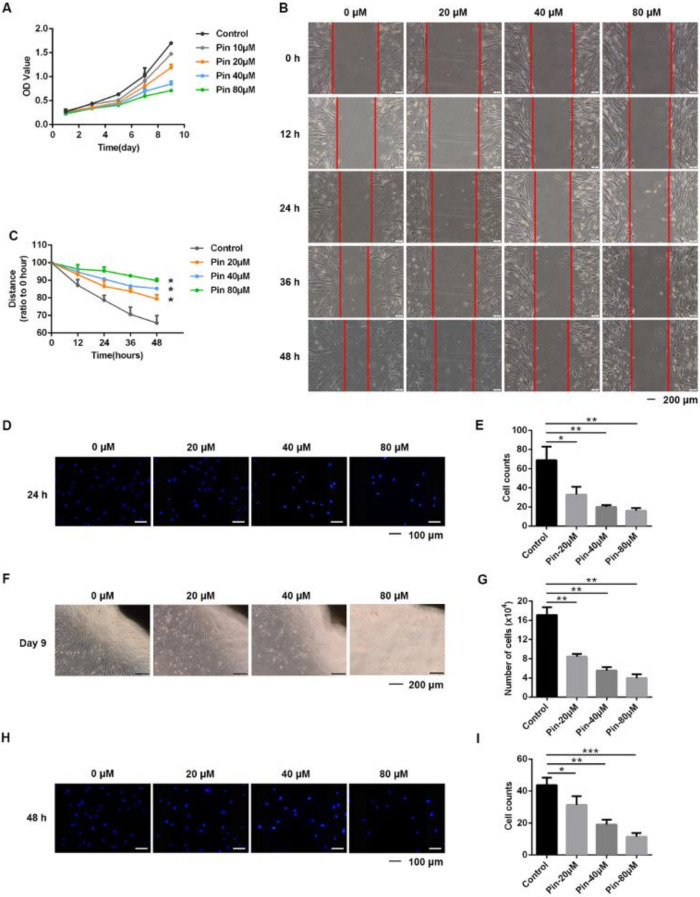
Pinocembrin suppresses the proliferation, migration, and invasion of keloid fibroblasts. (**A**) The CCK-8 assay of KFs. KFs were exposed to the indicated concentrations of pinocembrin (0 to 80 μM) for 1, 3, 5, 7, and 9 days. (**B**,**C**) The wound healing assay was used to analyze the migration of KFs treated with pinocembrin (0, 20, 40, 80 μM). The wound closure was photographed at 0, 12, 24, 36, and 48 h post-scratching. (**D**,**E**) Migratory capabilities of KFs treated with various concentrations of pinocembrin were investigated by transwell assays. Images were captured and counted under a fluorescence microscope at ×200 (scale bar = 100 μm). (**F**,**G**) Representative images of tissue explants cultured with pinocembrin at day 9 (×40, scale bar = 200 μm). The cell numbers that migrated out of the tissue explants were quantified at day 9. (**H**,**I**) Invasive capabilities of KFs treated with various concentrations of pinocembrin were investigated by transwell assays. Images were captured and counted under a fluorescence microscope at ×200 (scale bar = 100 μm). Data were presented as the means ± SD, n = 3. * *p* < 0.05; ** *p* < 0.01; *** *p* < 0.001.

**Figure 3 biomolecules-11-01240-f003:**
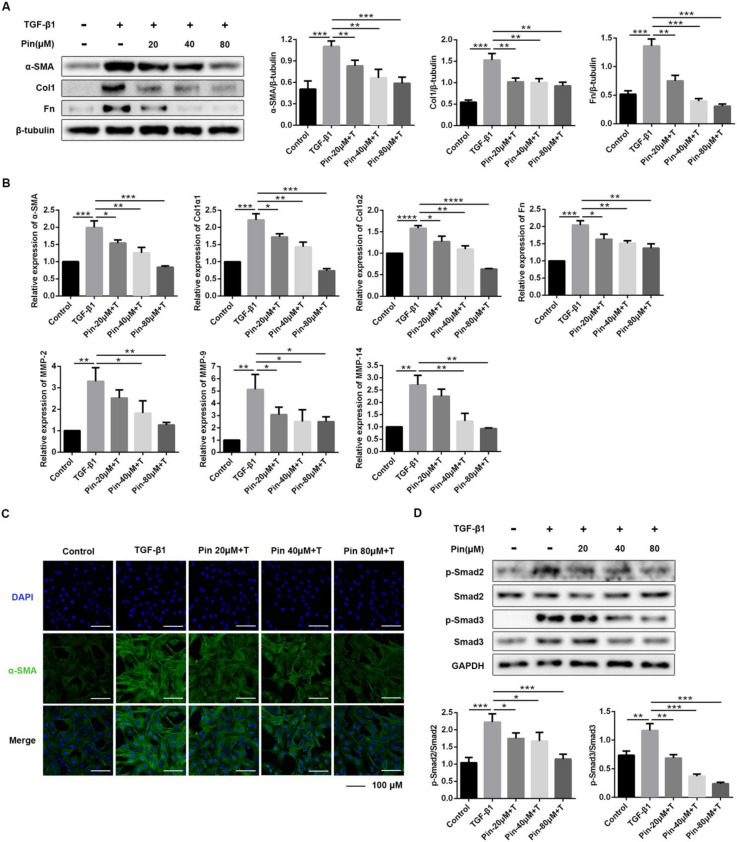
Pinocembrin attenuates TGF-β1-induced mouse primary dermal fibroblasts activation. (**A**) DFs were treated with TGF-β1 (5 ng·mL^−1^) and pinocembrin (0, 20, 40, 80 μM). (**A**) Western blot analysis of α-SMA, Col1, and Fn at 24 h post treatment. (**B**) qPCR analysis of α-SMA, Col1α1, Col1α2, Fn, MMP-2, MMP-9, and MMP-14 expression in DFs after 24 h of treatment. (**C**) Immunofluorescence analysis of α-SMA expression at 24 h post treatment (×400, scale bar = 100 μm). (**D**) The phosphorylation levels of Smad2 and Smad3 were analyzed by Western blot in KFs treated with TGF-β1 (5 ng·mL^−1^) and different doses of pinocembrin for 1 h. Data were presented as the means ± SD, n = 3. * *p* < 0.05; ** *p* < 0.01; *** *p* < 0.001; **** *p* < 0.0001.

**Figure 4 biomolecules-11-01240-f004:**
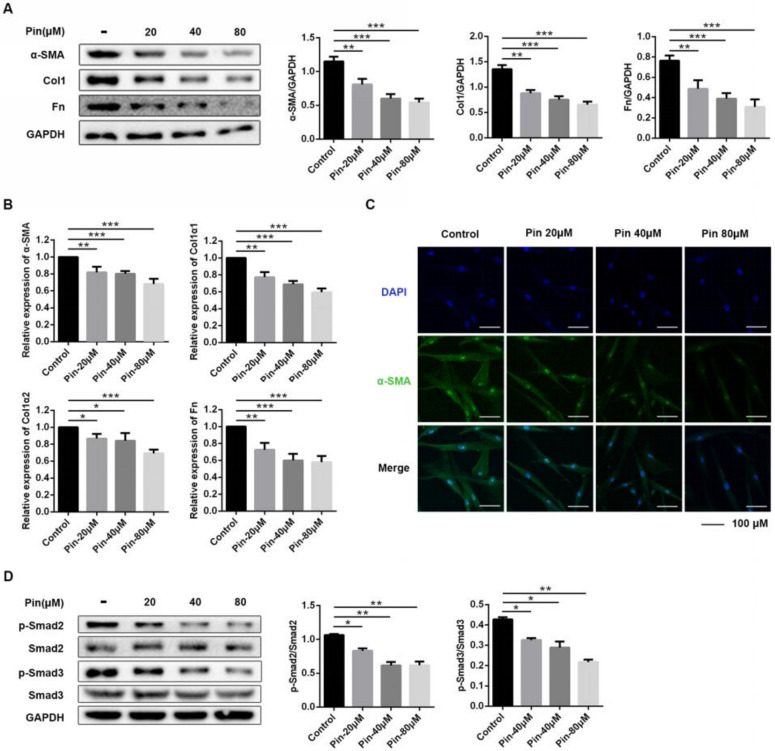
Pinocembrin reduces fibrosis-associated gene and protein expression in keloid fibroblasts. KFs were treated with pinocembrin (0, 20, 40, 80 μM) for 24 h. (**A**) The protein levels of α-SMA, Col1a1, Col1a2, and Fn were analyzed by Western blot. (**B**) The mRNA levels of α-SMA, Col1a1, Col1a2, and Fn were analyzed by quantitative real-time PCR. (**C**) Immunofluorescence staining of α-SMA was performed on KFs treated with pinocembrin (×400, scale bar = 100 μm). (**D**) The phosphorylation levels of Smad2 and Smad3 were analyzed by Western blot in KFs treated with pinocembrin. Data were presented as the means ± SD, n = 3. * *p* < 0.05; ** *p* < 0.01; *** *p* < 0.001.

**Figure 5 biomolecules-11-01240-f005:**
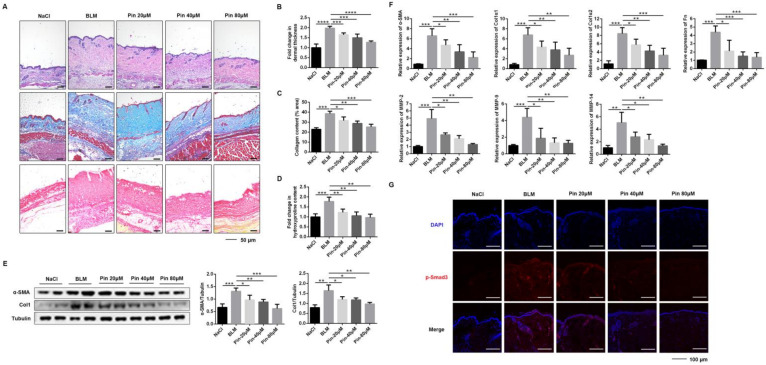
Pinocembrin ameliorates BLM-induced skin fibrosis in mice. (**A**) Representative skin sections stained with hematoxylin–eosin (H&E), Masson’s trichrome, and Sirius red staining (×100, Scale bar = 50 μm). (**B**) Total dermal thickness of the back of each group of mice. (**C**) Collagen density was quantified on Masson’s trichrome images. (**D**) Hydroxyproline content of skin tissues in mice. (**E**) Protein levels of α-SMA and Col1 were verified by Western blot in the lesional skin. (**F**) mRNA levels of α-SMA, Col1α1, Col1α2, Fn, MMP-2, MMP-9, and MMP-14 in the lesional skin were assessed by qPCR. (**G**) Immunofluorescence staining of p-Smad3 in skin frozen sections of BLM-induced model (×400, Scale bar = 100 μm). Means± SD, n = 8. * *p* < 0.05, ** *p* < 0.01, *** *p* < 0.001, **** *p* < 0.0001.

**Figure 6 biomolecules-11-01240-f006:**
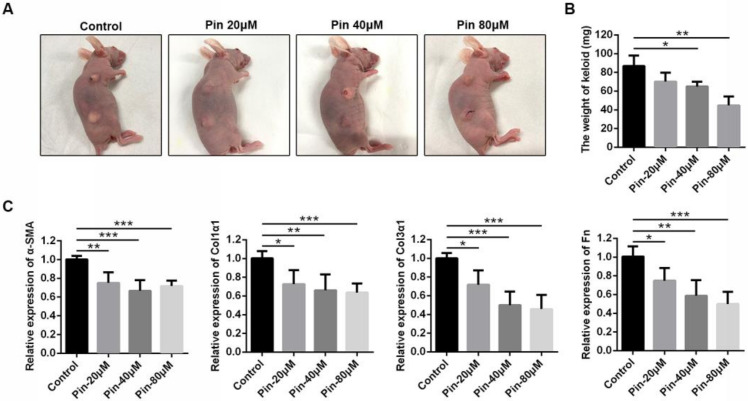
Pinocembrin accelerates the regression of xenografted keloid tissues and reduces ECM gene expression. (**A**) Macrographic examination of xenografted tissues on the back of nude BALB/c mice. (**B**) Weight of xenografted tissues after intralesional injection of pinocembrin. (**C**) mRNA levels of α-SMA, Col1α1, Col3α1, and Fn in xenografted keloid tissues were assessed by qPCR. Means± SD, n = 4. * *p* < 0.05, ** *p* < 0.01, *** *p* < 0.001.

**Table 1 biomolecules-11-01240-t001:** Specific primers used in real-time PCR analysis.

Gene	Primer	Sequence (5′–3′)
M-GAPDH	Forward	TGGATTTGGACGCATTGGTC
	Reverse	TTTGCACTGGTACGTGTTGAT
M-α-SMA	Forward	GCTGGTGATGATGCTCCCA
	Reverse	GCCCATTCCAACCATTACTCC
M-Col1α1	Forward	CCAAGAAGACATCCCTGAAGTCA
	Reverse	TGCACGTCATCGCACACA
M-Col1α2	Forward	GCAGGTTCACCTACTCTGTCCT
	Reverse	CTTGCCCCATTCATTTGTCT
M-Fn	Forward	AAGGATGGAGTGATAGCAACCC
	Reverse	TCTGCTTGAAATCTGGTGTGC
M-MMP-2	Forward	CAAGTTCCCCGGCGATGTC
	Reverse	TTCTGGTCAAGGTCACCTGTC
M-MMP-9	Forward	CTGGACAGCCAGACACTAAAG
	Reverse	CTCGCGGCAAGTCTTCAGAG
M-MMP-14	Forward	CAGTATGGCTACCTACCTCCAG
	Reverse	GCCTTGCCTGTCACTTGTAAA
H-β-actin	Forward	AGGCCAACCGTGAAAAGATG
	Reverse	AGAGCATAGCCCTCGTAGATGG
H-α-SMA	Forward	TGGGTGAACTCCATCGCTGTA
	Reverse	GTCGAATGCAACAAGGAAGCC
H-Col1α1	Forward	AAGCCGGAGGACAACCTTTTA
	Reverse	GCGAAGAGAATGACCAGATCC
H-Col1α2	Forward	GATGTTGAACTTGTTGCTGAGG
	Reverse	TCTTTCCCCATTCATTTGTCTT
H-Col3α1	Forward	TGGTGTTGGAGCCGCTGCCA
	Reverse	CTCAGCACTAGAATCTGTCC

Remarks: M, mouse; H, human.

## Data Availability

The data presented in this study are available on request from the corresponding author.

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
