# Peer review of "Pinocembrin Ameliorates Skin Fibrosis via Inhibiting TGF-β1 Signaling Pathway"

_biomolecules, 2021, doi:10.3390/biom11081240_

Round 1
Reviewer 1 Report
Dear Authors,
Thank you for your interesting and well-written paper.
I have some comments/questions for you:
- I miss a toxicity study for Pinocembrin. It could be possible that Pinocembrin is cell toxic and damages the cells, which in turn causes a stop of proliferation and differentiation.
- Using CCK-8 shows viability, however, under certain circumstances a higher signal does not mean necessary that there are more cells. Sometimes the reason is a faster metabolism or your cells gained more biomass or both. As example, 1000 fibroblast have a lower signal than 1000 myofibroblasts.
Therefore, a further cell counting method is needed and/or a microscopic control of the phenotype of treated/control cells.
- Migration assay: I would think that in the control wells are much more cells after 24-48 h. Thus, the probability that more cells migrate is increased. Please, check the ratio cells/migrated cells.And why using different incubation times 24-48 h?
- Unfortunately, your study is mostly descriptive (the reduction of pSMAS is only another read out) and you do not provide or elucidate a specific mechanism In my experience many kind of subtoxic stress can inhibit p-SMAD, a-SMA and proliferation
. So it is possible that for example:
-Pinocembrin is toxic/stressful and sends the cells into senescence-> Show that Pinocembrin is not toxic and did not induce cell stress (e.g. intracellular oxidative stress)
- Pinocembrin may scavenge/block important mediators/radicals/cytokines (not easy but please discuss)
You discussed that flavonoids with 5, 7, 3', 4'hydroxyl 342 substitutions could bind to activin receptor-like kinase 5 (ALK5) to down-regulate the signal transduction of TGF-β/Smads and reduce skin fibrosis.
Blocking TGFreceptor is quite interesting! Thus, ALK-5 inhibitors should have the same effects as Pinocembrin.
Why you did not test it with Pinocembrin?
Reviewer 2 Report
In this study the authors explored the potential effect and mechanisms of pinocembrin on skin fibrosis. This sudy showed that pinocembrin could suppress activation of fibroblasts by the inhibition of TGF-β1/Smad signaling and alleviate skin fibrosis. The study is well conducted with in vitro and in vivo experiments that support the conclusions.
Commets
The homeostase of the extracelular matrix (ECM) occurs, in skin, with collagen deposition but also with the degradation of ECM components, particularly by metaloproteinases (MMPs). TGF-β1/Smad signaling also regulates MMPs and the authors could evaluate how these protéases are involved in this model.
Reviewer 3 Report
The ms describes the protective role of pinocembrin on skin fibrosis.
Before acceptance, the authors need to formulate better some aspects of the works. Materials and methods should be better defined.
Why the authors choose these pinocembrin concentrations?
Conclusion section could be rephrased to put more emphasis on the possible applications of the study.
Round 2
Reviewer 1 Report
Dear Authors,
thank you for your revised manuscript.
Unfortunately, in my PDF version of your manuscript are some problems.
I cannot see the supplement figures. There are only in the response.Please check y-Axis viability!
Also you did not cite them in your manuscript only Figure S1. However, would integrate this Figure in the main manuscript.
Point 1/Point 2
You have used MTT assay on mouse dermal fibroblasts.
First, like cck-8 MTT measures metabolism, thus, the question whether proliferation is inhibited is not answered.
Second, you can not compare here fibroblast with keloid fibroblasts.
EG, the MTT assays showed that e.g. 80 µM pinocembrin did not have any effects after 24 h, whereas the cck-8 (Fig 1) you can see that there is a difference when you zoom in.
Also the figure 2 in the response letter shows that TGF increases metabolism, you need to check if therea are realy more cells (microscope).
Also the point if pinocembrin may be toxic is not clear. You show significant toxicity after 24 h with concentrations above 320 µM. Thus, you have to think that lower concentration may have also a slower poisining effect, which can be observed after a longer incubation time (48 h). Also, here it is important to differentiate between lower metabolism, toxic events (necrotic, apoptosis) and cell number.
Point 4/5
You have shown that Pinocembrin has strong toxic effects with 320 µM. lower concentrations can also induce subtoxic stress. Thus, the problem still remains, you need a prove if the ALK5 protein is involved and/or subtoxic stress may responsible for the observed effects.
Therefore, you have to discuss these critical points with more studies and literature within the introduction and discussion sections or you have to perform more experiments.
By the way, in all your PCR results you used normalized expression to the control. Please, use raw data. How you have calculated the SD and statistics for PCR data? Using the normalized means? Please indicate.
Author Response
Response to Reviewer 1 Comments
Point 1: Unfortunately, in my PDF version of your manuscript are some problems. I cannot see the supplement figures. There are only in the response. Please check y-Axis viability! Also, you did not cite them in your manuscript only Figure S1. However, would integrate this Figure in the main manuscript.
Response 1: We thank the reviewer’s suggestion. We have integrated the supplement figures in the main manuscript and modified the y-Axis viability.
Point 2: You have used MTT assay on mouse dermal fibroblasts.
First, like cck-8 MTT measures metabolism, thus, the question whether proliferation is inhibited is not answered.
Second, you cannot compare here fibroblast with keloid fibroblasts.
EG, the MTT assays showed that e.g. 80 µM pinocembrin did not have any effects after 24 h, whereas the cck-8 (Fig 1) you can see that there is a difference when you zoom in.
Also, the figure 2 in the response letter shows that TGF increases metabolism, you need to check if there are really more cells (microscope).
Also, the point if pinocembrin may be toxic is not clear. You show significant toxicity after 24 h with concentrations above 320 µM. Thus, you have to think that lower concentration may have also a slower poisining effect, which can be observed after a longer incubation time (48 h). Also, here it is important to differentiate between lower metabolism, toxic events (necrotic, apoptosis) and cell number.
Response 2: We thank the reviewer’s suggestion. First, we have supplemented the cell counting assay of mouse primary dermal fibroblasts and the results indicated that the cell number was significantly increased after 24/48 hours of growth and the cell number was more increased after stimulation with TGF-β1, as shown in Supplemental Figure 4.
Second, since the skin tissues of keloid patients has not been obtained recently, we cannot successfully supplement keloid fibroblasts related experiments. Because keloid fibroblasts are pathological activated skin fibroblasts, we use TGF-β1-treated mouse primary dermal fibroblasts to mimic the pathological activated skin fibroblasts. As shown in Supplemental Figure 3, MTT experiments indicated that pinocembrin has no effect on the proliferation of un-activated skin fibroblasts under normal conditions, but can significantly inhibit the proliferation of TGF-β1-activated skin fibroblasts. The CCK8 experiment was done using keloid fibroblasts, thus pinocembrin could inhibit the proliferation of pathological activated skin fibroblasts.
In addition, we also supplemented the flow cytometry test, and the results showed that the proportion of apoptotic and necrotic cells did not increase significantly after treated with pinocembrin (20 μM, 40 μM, 80 μM) for 48 h, which indicated that pinocembrin has no significant toxic effect on normal skin fibroblasts, as shown in Supplemental Figure 2.
Point 3: You have shown that Pinocembrin has strong toxic effects with 320 µM. lower concentrations can also induce subtoxic stress. Thus, the problem still remains, you need a prove if the ALK5 protein is involved and/or subtoxic stress may responsible for the observed effects. Therefore, you have to discuss these critical points with more studies and literature within the introduction and discussion sections or you have to perform more experiments.
Response 3: We thank the reviewer’s suggestion. We have supplemented the Flow cytometry assay of mouse primary dermal fibroblasts treated with pinocembrin, and the results showed that the effect of pinocembrin on apoptosis and necrosis of normal skin fibroblasts is not significant, as shown in Supplemental Figure 2.
Point 4: By the way, in all your PCR results you used normalized expression to the control. Please, use raw data. How you have calculated the SD and statistics for PCR data? Using the normalized means? Please indicate.
Response 4: We thank the reviewer’s comment. We have supplemented the calculation method of PCR results and cited references, as shown in the “2.9. Quantitative real-time PCR (qRT-PCR)” of “Materials and methods” section.